# Neural-Symbolic VQA: Disentangling Reasoning from Vision and Language Understanding

**Kexin Yi**[*]
Harvard University

**Jiajun Wu**[*]
MIT CSAIL

**Chuang Gan**
MIT-IBM Watson AI Lab

**Antonio Torralba**
MIT CSAIL

**Pushmeet Kohli**
DeepMind

**Joshua B. Tenenbaum**
MIT CSAIL

## Abstract

We marry two powerful ideas: deep representation learning for visual recognition and language understanding, and symbolic program execution for reasoning. Our neural-symbolic visual question answering (NS-VQA) system first recovers a structural scene representation from the image and a program trace from the question. It then executes the program on the scene representation to obtain an answer. Incorporating symbolic structure as prior knowledge offers three unique advantages. First, executing programs on a symbolic space is more robust to long program traces; our model can solve complex reasoning tasks better, achieving an accuracy of 99.8% on the CLEVR dataset. Second, the model is more data- and memory-efficient: it performs well after learning on a small number of training data; it can also encode an image into a compact representation, requiring less storage than existing methods for offline question answering. Third, symbolic program execution offers full transparency to the reasoning process; we are thus able to interpret and diagnose each execution step.

## 1 Introduction

Looking at the images and questions in Figure 1, we instantly recognize objects and their attributes, parse complicated questions, and leverage such knowledge to reason and answer the questions. We can also clearly explain how we reason to obtain the answer. Now imagine that you are standing in front of the scene, eyes closed, only able to build your scene representation through touch. Not surprisingly, reasoning without vision remains effortless. For humans, reasoning is fully interpretable, and not necessarily interwoven with visual perception.

The advances in deep representation learning and the development of large-scale datasets [Malinowski and Fritz, 2014, Antol et al., 2015] have inspired a number of pioneering approaches in visual question-answering (VQA), most trained in an end-to-end fashion [Yang et al., 2016]. Though innovative, pure neural net–based approaches often perform less well on challenging reasoning tasks. In particular, a recent study [Johnson et al., 2017a] designed a new VQA dataset, CLEVR, in which each image comes with intricate, compositional questions generated by programs, and showed that state-of-the-art VQA models did not perform well.

Later, Johnson et al. [2017b] demonstrated that machines can learn to reason by wiring in prior knowledge of human language as programs. Specifically, their model integrates a program generator that infers the underlying program from a question, and a learned, attention-based executor that runs the program on the input image. Such a combination achieves very good performance on the CLEVR

---

[*] indicates equal contributions. Project page: `http://nsvqa.csail.mit.edu`

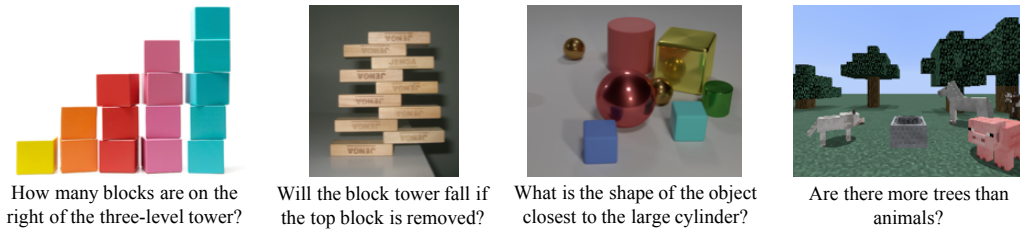

<div style="text-align:center">

How many blocks are on the right of the three-level tower?    Will the block tower fall if the top block is removed?    What is the shape of the object closest to the large cylinder?    Are there more trees than animals?

</div>

Figure 1: Human reasoning is interpretable and disentangled: we first draw abstract knowledge of the scene via visual perception and then perform logic reasoning on it. This enables compositional, accurate, and generalizable reasoning in rich visual contexts.

dataset, and generalizes reasonably well to CLEVR-Humans, a dataset that contains the same images as CLEVR but now paired with human-generated questions. However, their model still suffers from two limitations: first, training the program generator requires many annotated examples; second, the behaviors of the attention-based neural executor are hard to explain. In contrast, we humans can reason on CLEVR and CLEVR-Humans even with a few labeled instances, and we can also clearly explain how we do it.

In this paper, we move one step further along the spectrum of learning vs. modeling, proposing a neural-symbolic approach for visual question answering (NS-VQA) that fully disentangles vision and language understanding from reasoning. We use neural networks as powerful tools for parsing— inferring structural, object-based scene representation from images, and generating programs from questions. We then incorporate a symbolic program executor that, complementary to the neural parser, runs the program on the scene representation to obtain an answer.

The combination of deep recognition modules and a symbolic program executor offers three unique advantages. First, the use of symbolic representation offers robustness to long, complex program traces. It also reduces the need of training data. On the CLEVR dataset, our method is trained on questions with 270 program annotations plus 4K images, and is able to achieve a near-perfect accuracy of 99.8%.

Second, both our reasoning module and visual scene representation are light-weighted, requiring minimal computational and memory cost. In particular, our compact structural image representation requires much less storage during reasoning, reducing the memory cost by 99% compared with other state-of-the-art algorithms.

Third, the use of symbolic scene representation and program traces forces the model to accurately recover underlying programs from questions. Together with the fully transparent and interpretable nature of symbolic representations, the reasoning process can be analyzed and diagnosed step-by-step.

## 2 Related Work

**Structural scene representation.** Our work is closely related to research on learning an interpretable, disentangled representation with a neural network [Kulkarni et al., 2015, Yang et al., 2015, Wu et al., 2017]. For example, Kulkarni et al. [2015] proposed convolutional inverse graphics networks that learn to infer the pose and lighting of a face; Yang et al. [2015] explored learning disentangled representations of pose and content from chair images. There has also been work on learning disentangled representation without direct supervision [Higgins et al., 2018, Siddharth et al., 2017, Vedantam et al., 2018], some with sequential generative models [Eslami et al., 2016, Ba et al., 2015]. In a broader view, our model also relates to the field of "vision as inverse graphics" [Yuille and Kersten, 2006]. Our NS-VQA model builds upon the structural scene representation [Wu et al., 2017] and explores how it can be used for visual reasoning.

**Program induction from language.** Recent papers have explored using program search and neural networks to recover programs from a domain-specific language [Balog et al., 2017, Neelakantan et al., 2016, Parisotto et al., 2017]. For sentences, semantic parsing methods map them to logical forms via a knowledge base or a program [Berant et al., 2013, Liang et al., 2013, Vinyals et al., 2015, Guu et al., 2017]. In particular, Andreas et al. [2016] attempted to use the latent structure in

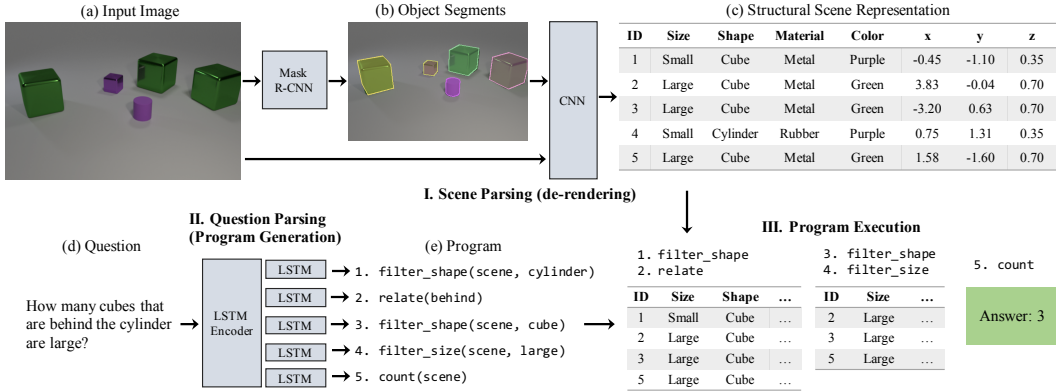

Figure 2: Our model has three components: first, a scene parser (de-renderer) that segments an input image (a-b) and recovers a structural scene representation (c); second, a question parser (program generator) that converts a question in natural language (d) into a program (e); third, a program executor that runs the program on the structural scene representation to obtain the answer.

language to help question answering and reasoning, Rothe et al. [2017] studied the use of formal programs in modeling human questions, and Goldman et al. [2018] used abstract examples to build weakly-supervised semantic parsers.

**Visual question answering.**   Visual question answering (VQA) [Malinowski and Fritz, 2014, Antol et al., 2015] is a versatile and challenging test bed for AI systems. Compared with the well-studied text-based question answering, VQA emerges by its requirement on both semantic and visual understanding. There have been numerous papers on VQA, among which some explicitly used structural knowledge to help reasoning [Wang et al., 2017]. Current leading approaches are based on neural attentions [Yang et al., 2016, Lu et al., 2016], which draw inspiration from human perception and learn to attend the visual components that serve as informative evidence to the question. Nonetheless, Jabri et al. [2016] recently proposed a remarkably simple yet effective classification baseline. Their system directly extracts visual and text features from *whole* images and questions, concatenates them, and trains multi-class classifiers to select answers. This paper, among others [Goyal et al., 2017], reveals potential caveats in the proposed VQA systems—models are overfitting dataset biases.

**Visual reasoning.**   Johnson et al. [2017a] built a new VQA dataset, named CLEVR, carefully controlling the potential bias and benchmarking how well models reason. Their subsequent model achieved good results on CLEVR by combining a recurrent program generator and an attentive execution engine [Johnson et al., 2017b]. There have been other end-to-end neural models that has achieved nice performance on the dataset, exploiting various attention structures and accounting object relations [Hudson and Manning, 2018, Santoro et al., 2017, Hu et al., 2017, Perez et al., 2018, Zhu et al., 2017]. More recently, several papers have proposed to directly incorporate the syntactic and logic structures of the reasoning task to the attentive module network's architecture for reasoning. These structures include the underlying functional programs [Mascharka et al., 2018, Suarez et al., 2018] and dependency trees [Cao et al., 2018] of the input question. However, training of the models relies heavily on these extra signals.

From a broader perspective, Misra et al. [2018] explored learning to reason by asking questions and Bisk et al. [2018] studied spatial reasoning in a 3D blocks world. Recently, Aditya et al. [2018] incorporated probabilistic soft logic into a neural attention module and obtained some interpretability of the model, and Gan et al. [2017] learned to associate image segments with questions. Our model moves along this direction further by modeling the entire scene into an object-based, structural representation, and integrating it with a fully transparent and interpretable symbolic program executor.

## 3 Approach

Our NS-VQA model has three components: a scene parser (de-renderer), a question parser (program generator), and a program executor. Given an image-question pair, the scene parser de-renders the image to obtain a structural scene representation (Figure 2-I), the question parser generates a hierarchical program from the question (Figure 2-II), and the executor runs the program on the structural representation to obtain an answer (Figure 2-III).

Our scene parser recovers a structural and disentangled representation of the scene in the image (Figure 2a), based on which we can perform fully interpretable symbolic reasoning. The parser takes a two-step, segment-based approach for de-rendering: it first generates a number of segment proposals (Figure 2b), and for each segment, classifies the object and its attributes. The final, structural scene representation is disentangled, compact, and rich (Figure 2c).

The question parser maps an input question in natural language (Figure 2d) to a latent program (Figure 2e). The program has a hierarchy of functional modules, each fulfilling an independent operation on the scene representation. Using a hierarchical program as our reasoning backbone naturally supplies compositionality and generalization power.

The program executor takes the output sequence from the question parser, applies these functional modules on the abstract scene representation of the input image, and generates the final answer (Figure 2-III). The executable program performs purely symbolic operations on its input throughout the entire execution process, and is fully deterministic, disentangled, and interpretable with respect to the program sequence.

### 3.1 Model Details

**Scene parser.** For each image, we use Mask R-CNN [He et al., 2017] to generate segment proposals of all objects. Along with the segmentation mask, the network also predicts the categorical labels of discrete intrinsic attributes such as color, material, size, and shape. Proposals with bounding box score less than 0.9 are dropped. The segment for each single object is then paired with the original image, resized to 224 by 224 and sent to a ResNet-34 [He et al., 2015] to extract the spacial attributes such as pose and 3D coordinates. Here the inclusion of the original full image enables the use of contextual information.

**Question parser.** Our question parser is an attention-based sequence to sequence (seq2seq) model with an encoder-decoder structure similar to that in Luong et al. [2015] and Bahdanau et al. [2015]. The encoder is a bidirectional LSTM [Hochreiter and Schmidhuber, 1997] that takes as input a question of variable lengths and outputs an encoded vector $e_i$ at time step $i$ as

$$e_i = [e_i^F, e_i^B], \quad \text{where} \quad e_i^F, h_i^F = \text{LSTM}(\Phi_E(x_i), h_{i-1}^F), \quad e_i^B, h_i^B = \text{LSTM}(\Phi_E(x_i), h_{i+1}^B). \tag{1}$$

Here $\Phi_E$ is the jointly trained encoder word embedding. $(e_i^F, h_i^F)$, $(e_i^B, h_i^B)$ are the outputs and hidden vectors of the forward and backward networks at time step $i$. The decoder is a similar LSTM that generates a vector $q_t$ from the previous token of the output sequence $y_{t-1}$. $q_t$ is then fed to an attention layer to obtain a context vector $c_t$ as a weighted sum of the encoded states via

$$q_t = \text{LSTM}(\Phi_D(y_{t-1})), \qquad \alpha_{ti} \propto \exp(q_t^\top W_A e_i), \qquad c_t = \sum_i \alpha_{ti} e_i. \tag{2}$$

$\Phi_D$ is the decoder word embedding. For simplicity we set the dimensions of vectors $q_t, e_i$ to be the same and let the attention weight matrix $W_A$ to be an identity matrix. Finally, the context vector, together with the decoder output, is passed to a fully connected layer with softmax activation to obtain the distribution for the predicted token $y_t \sim \text{softmax}(W_O[q_t, c_t])$. Both the encoder and decoder have two hidden layers with a 256-dim hidden vector. We set the dimensions of both the encoder and decoder word vectors to be 300.

**Program executor.** We implement the program executor as a collection of deterministic, generic functional modules in Python, designed to host all logic operations behind the questions in the dataset. Each functional module is in one-to-one correspondence with tokens from the input program sequence, which has the same representation as in Johnson et al. [2017b]. The modules share the same input/output interface, and therefore can be arranged in any length and order. A typical program

| Methods | Count | Exist | Compare Number | Compare Attribute | Query Attribute | Overall |
|---|---|---|---|---|---|---|
| Humans [Johnson et al., 2017b] | 86.7 | 96.6 | 86.4 | 96.0 | 95.0 | 92.6 |
| CNN+LSTM+SAN [Johnson et al., 2017b] | 59.7 | 77.9 | 75.1 | 70.8 | 80.9 | 73.2 |
| N2NMN* [Hu et al., 2017] | 68.5 | 85.7 | 84.9 | 88.7 | 90.0 | 83.7 |
| Dependency Tree [Cao et al., 2018] | 81.4 | 94.2 | 81.6 | 97.1 | 90.5 | 89.3 |
| CNN+LSTM+RN [Santoro et al., 2017] | 90.1 | 97.8 | 93.6 | 97.1 | 97.9 | 95.5 |
| IEP* [Johnson et al., 2017b] | 92.7 | 97.1 | 98.7 | 98.9 | 98.1 | 96.9 |
| CNN+GRU+FiLM [Perez et al., 2018] | 94.5 | 99.2 | 93.8 | 99.0 | 99.2 | 97.6 |
| DDRprog* [Suarez et al., 2018] | 96.5 | 98.8 | 98.4 | 99.0 | 99.1 | 98.3 |
| MAC [Hudson and Manning, 2018] | 97.1 | 99.5 | 99.1 | 99.5 | 99.5 | 98.9 |
| TbD+reg+hres* [Mascharka et al., 2018] | 97.6 | 99.2 | 99.4 | 99.6 | 99.5 | 99.1 |
| NS-VQA (ours, 90 programs) | 64.5 | 87.4 | 53.7 | 77.4 | 79.7 | 74.4 |
| NS-VQA (ours, 180 programs) | 85.0 | 92.9 | 83.4 | 90.6 | 92.2 | 89.5 |
| NS-VQA (ours, 270 programs) | **99.7** | **99.9** | **99.9** | **99.8** | **99.8** | **99.8** |

Table 1: Our model (NS-VQA) outperforms current state-of-the-art methods on CLEVR and achieves near-perfect question answering accuracy. The question-program pairs used for pretraining our model are uniformly drawn from the 90 question families of the dataset: 90, 180, 270 programs correspond to 1, 2, 3 samples from each family respectively. (*): trains on all program annotations (700K).

sequence begins with a `scene` token, which signals the input of the original scene representation. Each functional module then sequentially executes on the output of the previous one. The last module outputs the final answer to the question. When type mismatch occurs between input and output across adjacent modules, an `error` flag is raised to the output, in which case the model will randomly sample an answer from all possible outputs of the final module. Figure 3 shows two examples.

### 3.2 Training Paradigm

**Scene parsing.** Our implementation of the object proposal network (Mask R-CNN) is based on "Detectron" [Girshick et al., 2018]. We use ResNet-50 FPN [Lin et al., 2017] as the backbone and train the model for 30,000 iterations with eight images per batch. Please refer to He et al. [2017] and Girshick et al. [2018] for more details. Our feature extraction network outputs the values of continuous attributes. We train the network on the *proposed* object segments computed from the training data using the mean square error as loss function for 30,000 iterations with learning rate 0.002 and batch size 50. Both networks of our scene parser are trained on 4,000 generated CLEVR images.

**Reasoning.** We adopt the following two-step procedure to train the question parser to learn the mapping from a question to a program. First, we select a small number of ground truth question-program pairs from the training set to pretrain the model with direct supervision. Then, we pair it with our deterministic program executor, and use REINFORCE [Williams, 1992] to fine-tune the parser on a larger set of question-answer pairs, using only the correctness of the execution result as the reward signal.

During supervised pretraining, we train with learning rate $7 \times 10^{-4}$ for 20,000 iterations. For reinforce, we set the learning rate to be $10^{-5}$ and run at most 2M iterations with early stopping. The reward is maximized over a constant baseline with a decay weight 0.9 to reduce variance. Batch size is fixed to be 64 for both training stages. All our models are implemented in PyTorch.

## 4 Evaluations

We demonstrate the following advantages of our disentangled structural scene representation and symbolic execution engine. First, our model can learn from a small number of training data and outperform the current state-of-the-art methods while precisely recovering the latent programs (Sections 4.1). Second, our model generalizes well to other question styles (Sections 4.3), attribute combinations (Sections 4.2), and visual context (Section 4.4). Code of our model is available at https://github.com/kexinyi/ns-vqa

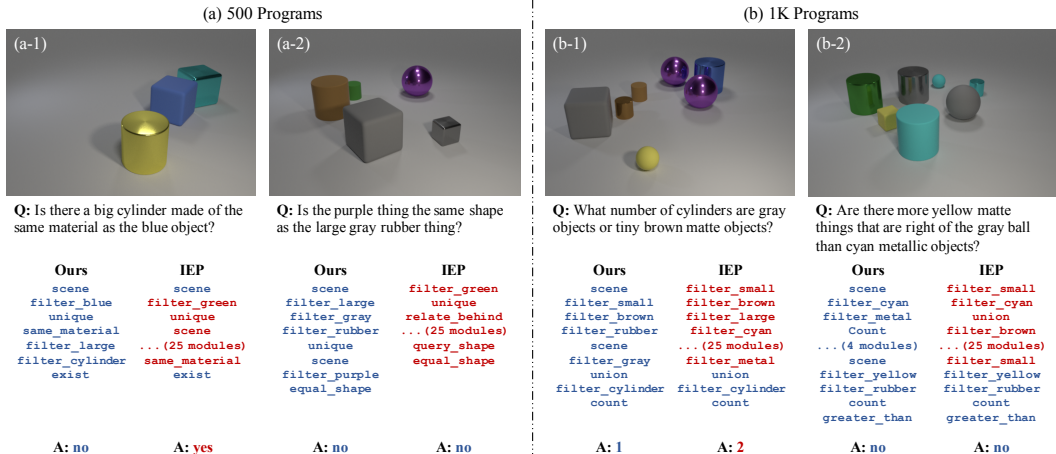

Figure 3: Qualitative results on CLEVR. Blue color indicates correct program modules and answers; red indicates wrong ones. Our model is able to robustly recover the correct programs compared to the IEP baseline.

## 4.1 Data-Efficient, Interpretable Reasoning

**Setup.** We evaluate our NS-VQA on CLEVR [Johnson et al., 2017a]. The dataset includes synthetic images of 3D primitives with multiple attributes—shape, color, material, size, and 3D coordinates. Each image has a set of questions, each of which associates with a program (a set of symbolic modules) generated by machines based on 90 logic templates.

Our structural scene representation for a CLEVR image characterizes the objects in it, each labeled with its shape, size, color, material, and 3D coordinates (see Figure 2c). We evaluate our model's performance on the validation set under various supervise signal for training, including the numbers of ground-truth programs used for pretraining and question-answer pairs for REINFORCE. Results are compared with other state-of-the-art methods including the IEP baseline [Johnson et al., 2017b]. We not only assess the correctness of the answer obtained by our model, but also how well it recovers the underlying program. An interpretable model should be able to output the correct program in addition to the correct answer.

**Results.** Quantitative results on the CLEVR dataset are summarized in Table 1. Our NS-VQA achieves near-perfect accuracy and outperforms other methods on all five question types. We first pretrain the question parser on 270 annotated programs sampled across the 90 question templates (3 questions per template), a number below the weakly supervised limit suggested by Johnson et al. [2017b] (9K), and then run REINFORCE on all the question-answer pairs. Repeated experiments starting from different sets of programs show a standard deviation of less than 0.1 percent on the results for 270 pretraining programs (and beyond). The variances are larger when we train our model with fewer programs (90 and 180). The reported numbers are the mean of three runs.

We further investigate the data-efficiency of our method with respect to both the number of programs used for pretraining and the overall question-answer pairs used in REINFORCE. Figure 4a shows the result when we vary the number of pretraining programs. NS-VQA outperforms the IEP baseline under various conditions, even with a weaker supervision during REINFORCE (2K and 9K question-answer pairs in REINFORCE). The number of question-answer pairs can be further reduced by pretraining the model on a larger set of annotated programs. For example, our model achieves the same near-perfect accuracy of 99.8% with 9K question-answer pairs with annotated programs for both pretraining and REINFORCE.

Figure 4b compares how well our NS-VQA recovers the underlying programs compared to the IEP model. IEP starts to capture the true programs when trained with over 1K programs, and only recovers half of the programs with 9K programs. Qualitative examples in Figure 3 demonstrate that IEP tends to fake a long wrong program that leads to the correct answer. In contrast, our model achieves 88%

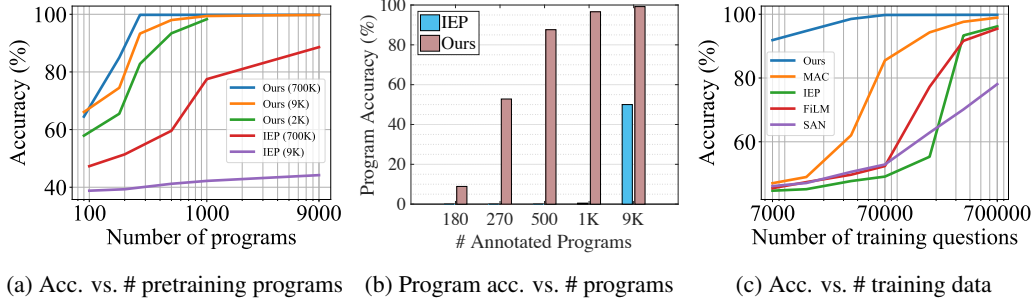

(a) Acc. vs. # pretraining programs    (b) Program acc. vs. # programs    (c) Acc. vs. # training data

Figure 4: Our model exhibits high data efficiency while achieving state-of-the-art performance and preserving interpretability. (a) QA accuracy vs. number of programs used for pretraining; different curves indicate different numbers of question-answer pairs used in the REINFORCE stage. (c) QA accuracy vs. total number of training question-answer pairs; our model is pretrained on 270 programs.

program accuracy with 500 annotations, and performs almost perfectly on both question answering and program recovery with 9K programs.

Figure 4c shows the QA accuracy vs. the number of questions and answers used for training, where our NS-VQA has the highest performance under all conditions. Among the baseline methods we compare with, MAC [Hudson and Manning, 2018] obtains high accuracy with zero program annotations; in comparison, our method needs to be pretrained on 270 program annotations, but requires fewer question-answer pairs to reach similar performance.

Our model also requires minimal memory for offline question answering: the structural representation of each image only occupies less than 100 bytes; in comparison, attention-based methods like IEP requires storing either the original image or its feature maps, taking at least 20K bytes per image.

## 4.2 Generalizing to Unseen Attribute Combinations

Recent neural reasoning models have achieved impressive performance on the original CLEVR QA task [Johnson et al., 2017b, Mascharka et al., 2018, Perez et al., 2018], but they generalize less well across biased dataset splits. This is revealed on the CLEVR-CoGenT dataset [Johnson et al., 2017a], a benchmark designed specifically for testing models' generalization to novel attribute compositions.

**Setup.** The CLEVR-CoGenT dataset is derived from CLEVR and separated into two biased splits: split A only contains cubes that are either gray, blue, brown or yellow, and cylinders that are red, green, purple or cyan; split B has the opposite color-shape pairs for cubes and cylinders. Both splits contain spheres of any color. Split A has 70K images and 700K questions for training and both splits have 15K images and 150K questions for evaluation and testing. The desired behavior of a generalizable model is to perform equally well on both splits while only trained on split A.

**Results.** Table 2a shows the generalization results with a few interesting findings. The vanilla NS-VQA trained purely on split A and fine-tuned purely on split B (1000 images) does not generalize as well as the state-of-the-art. We observe that this is because of the bias in the attribute recognition network of the scene parser, which learns to classify object shape based on color. NS-VQA works well after we fine-tune it on data from both splits (4000 A, 1000 B). Here, we only fine-tune the attribute recognition network with annotated images from split B, but no questions or programs; thanks to the disentangled pipeline and symbolic scene representation, our question parser and executor are not overfitting to particular splits. To validate this, we train a separate shape recognition network that takes gray-scale but not color images as input (NS-VQA+Gray). The augmented model works well on both splits without seeing any data from split B. Further, with an image parser trained on the original condition (i.e. the same as in CLEVR), our question parser and executor also generalize well across splits (NS-VQA+Ori).

| Methods | Not Fine-tuned | | Fine-tune on | Fine-tuned | |
|---|---|---|---|---|---|
| | A | B | | A | B |
| CNN+LSTM+SA | 80.3 | 68.7 | B | 75.7 | 75.8 |
| IEP (18K programs) | 96.6 | 73.7 | B | 76.1 | 92.7 |
| CNN+GRU+FiLM | 98.3 | 78.8 | B | 81.1 | 96.9 |
| TbD+reg | 98.8 | 75.4 | B | 96.9 | 96.3 |
| NS-VQA (ours) | **99.8** | 63.9 | B | 64.9 | 98.9 |
| NS-VQA (ours) | **99.8** | 63.9 | A+B | **99.6** | **99.0** |
| NS-VQA+Gray (ours) | 99.6 | 98.4 | - | - | - |
| NS-VQA+Ori (ours) | **99.8** | **99.7** | - | - | - |

(a) Generalization results on CLEVR-CoGenT.

| # Programs | NS-VQA | IEP |
|---|---|---|
| 100 | **60.2** | 38.7 |
| 200 | **65.2** | 40.1 |
| 500 | **67.8** | 49.2 |
| 1K | **67.8** | 63.4 |
| 18K | **67.0** | 66.6 |

(b) Question answering accuracy on CLEVR-Humans.

Table 2: Generalizing to unseen attribute compositions and question styles. (a) Our image parser is trained on 4,000 synthetic images from split A and fine-tuned on 1,000 images from split B. The question parser is *only* trained on split A starting from 500 programs. Baseline methods are fine-tuned on 3K images plus 30K questions from split B. NS-VQA+Gray adopts a gray channel in the image parser for shape recognition and NS-VQA+Ori uses an image parser trained from the original images from CLEVR. Please see text for more details. (b) Our model outperforms IEP on CLEVR-Humans under various training conditions.

## 4.3 Generalizing to Questions from Humans

Our model also enables efficient generalization toward more realistic question styles over the same logic domain. We evaluate this on the CLEVR-Humans dataset, which includes human-generated questions on CLEVR images (see Johnson et al. [2017b] for details). The questions follow real-life human conversation style without a regular structural expression.

**Setup.** We adopt a training paradigm for CLEVR-Humans similar to the original CLEVR dataset: we first pretrain the model with a limited number of programs from CLEVR, and then fine-tune it on CLEVR-Humans with REINFORCE. We initialize the encoder word embedding by the GloVe word vectors [Pennington et al., 2014] and keep it fixed during pretraining. The REINFORCE stage lasts for at most 1M iterations; early stop is applied.

**Results.** The results on CLEVR-Humans are summarized in Table 2b. Our NS-VQA outperforms IEP on CLEVR-Humans by a considerable margin under small amount of annotated programs. This shows our structural scene representation and symbolic program executor helps to exploit the strong exploration power of REINFORCE, and also demonstrates the model's generalizability across different question styles.

## 4.4 Extending to New Scene Context

Structural scene representation and symbolic programs can also be extended to other visual and contextual scenarios. Here we show results on reasoning tasks from the Minecraft world.

**Setup.** We now consider a new dataset where objects and scenes are taken from Minecraft and therefore have drastically different scene context and visual appearance. We use the dataset generation tool provided by Wu et al. [2017] to render 10,000 Minecraft scenes, building upon the Malmo interface [Johnson et al., 2016]. Each image consists of 3 to 6 objects, and each object is sampled from a set of 12 entities. We use the same configuration details as suggested by Wu et al. [2017]. Our structural representation has the following fields for each object: category (12-dim), position in the 2D plane (2-dim, $\{x, z\}$), and the direction the object faces {front, back, left, right}. Each object is thus encoded as a 18-dim vector.

We generate diverse questions and programs associated with each Minecraft image based on the objects' categorical and spatial attributes (position, direction). Each question is composed as a hierarchy of three families of basic questions: first, querying object attributes (class, location, direction); second, counting the number of objects satisfying certain constraints; third, verifying if

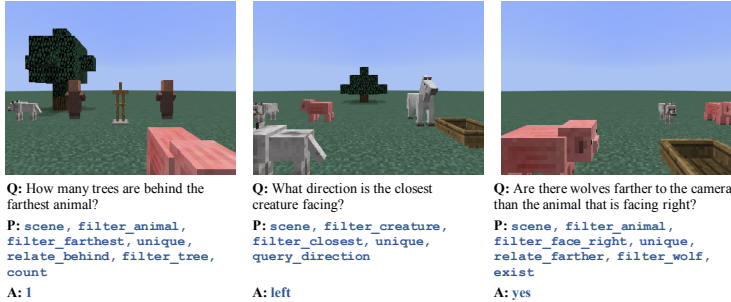

| # Programs | Accuracy |
|---|---|
| 50 | 71.1 |
| 100 | 72.4 |
| 200 | 86.9 |
| 500 | 87.3 |

(a) Sample results on the Minecraft dataset.

(b) Question answering accuracy with different numbers of annotated programs

Figure 5: Our model also applies to Minecraft, a world with rich and hierarchical scene context and different visual appearance.

an object has certain property. Our dataset differs from CLEVR primarily in two ways: Minecraft hosts a larger set of 3D objects with richer image content and visual appearance; our questions and programs involve hierarchical attributes. For example, a "wolf" and a "pig" are both "animals", and an "animal" and a "tree" are both "creatures". We use the first 9,000 images with 88,109 questions for training and the remaining 1,000 images with 9,761 questions for testing. We follow the same recipe as described in Section 3.2 for training on Minecraft.

**Results.** Quantitative results are summarized in Table 5b. The overall behavior is similar to that on the CLEVR dataset, except that reasoning on Minecraft generally requires weaker initial program signals. Figure 5a shows the results on three test images: our NS-VQA finds the correct answer and recovers the correct program under the new scene context. Also, most of our model's wrong answers on this dataset are due to errors in perceiving heavily occluded objects, while the question parser still preserves its power to parse input questions.

# 5 Discussion

We have presented a neural-symbolic VQA approach that disentangles reasoning from visual perception and language understanding. Our model uses deep learning for inverse graphics and inverse language modeling—recognizing and characterizing objects in the scene; it then uses a symbolic program executor to reason and answer questions.

We see our research suggesting a possible direction to unify two powerful ideas: deep representation learning and symbolic program execution. Our model connects to, but also differs from the recent pure deep learning approaches for visual reasoning. Wiring in symbolic representation as prior knowledge increases performance, reduces the need for annotated data and for memory significantly, and makes reasoning fully interpretable.

The machine learning community has often been skeptical of symbolic reasoning, as symbolic approaches can be brittle or have difficulty generalizing to natural situations. Some of these concerns are less applicable to our work, as we leverage learned abstract representations for mapping both visual and language inputs to an underlying symbolic reasoning substrate. However, building structured representations for scenes and sentence meanings—the targets of these mappings—in ways that generalize to truly novel situations remains a challenge for many approaches including ours. Recent progress on unsupervised or weakly supervised representation learning, in both language and vision, offers some promise of generalization. Integrating this work with our neural-symbolic approach to visually grounded language is a promising future direction.

#### Acknowledgments

We thank Jiayuan Mao, Karthik Narasimhan, and Jon Gauthier for helpful discussions and suggestions. We also thank Drew A. Hudson for sharing experimental results for comparison. This work is in part supported by ONR MURI N00014-16-1-2007, the Center for Brain, Minds, and Machines (CBMM), IBM Research, and Facebook.

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
