[Supplementary Material · supplementary.pdf]

# Supplementary Material

## A   Scene Parser Details

**Data.**   Our scene parser is trained on 4,000 CLEVR-style images rendered by Blender with object masks and ground-truth attributes including color, material, shape, size, and 3D coordinates. Because the original CLEVR dataset does not include object masks, we generate these 4,000 training images ourselves using the CLEVR dataset generation tool[*]. For the CLEVR-CoGenT experiment, we generate another set of images that satisfy the attribute composition restrictions, using the same software.

**Training.**   We first train the Mask-RCNN object detector on the rendered images and masks. For the CLEVR dataset, the bounding-box classifier contains 48 classes, each representing one composition of object intrinsic attributes of three shapes, two materials, and eight colors (i.e. "blue rubber cube"). Then we run the detector on the same training images to obtain object segmentation proposals, and pair each segment to a labeled object. The segment-label pairs are then used for training the feature extraction CNN. Before entering the CNN, the object segment is concatenated with the original image to provide contextual information.

## B   Program Executor Details

Our program executor is implemented as a collection of functional modules in Python, each executing a designated logic operation on a abstract scene representation. Given a program sequence, the modules are executed one by one; The output of a module is iteratively passed to the next. The input and output types of the modules include the following: *object*, a dictionary containing the full abstract representation of a single object; *scene*, a list of objects; *entry*, an indicator of any object attribute values (color, material, shape, size); *number*; *boolean*. All program modules are summarized in the following tables.

| Module | Input type | Output type | Description |
|---|---|---|---|
| `scene` | - | *scene* | Return a list of all objects |
| `unique` | *scene* | *object* | Return the only object in the scene |
| `union` | *scene* | *scene* | Return the union of two scenes |
| `intersect` | *scene* | *scene* | Return the intersection of two scenes |
| `count` | *scene* | *number* | Return the number of objects in a scene |

Table 1: Set operation modules of the program executor.

| Module | Input type | Output type | Description |
|---|---|---|---|
| `equal_color` | (*entry*, *entry*) | *Boolean* | Return whether input colors are the same |
| `equal_material` | (*entry*, *entry*) | *Boolean* | Return whether input materials are the same |
| `equal_shape` | (*entry*, *entry*) | *Boolean* | Return whether input shapes are the same |
| `equal_size` | (*entry*, *entry*) | *Boolean* | Return whether input sizes are the same |
| `equal_integer` | (*number*, *number*) | *Boolean* | Return whether input numbers equal |
| `greater_than` | (*number*, *number*) | *Boolean* | Return whether the first number is greater than the second |
| `less_than` | (*number*, *number*) | *Boolean* | Return whether the first number is less than the second |
| `exist` | *scene* | *Boolean* | Return whether the input scene includes any object |

Table 2: Boolean operation modules of the program executor.

---

[*]https://github.com/facebookresearch/clevr-dataset-gen

| Module | Input type | Output type | Description |
|---|---|---|---|
| query_color | *object* | *entry* | Return the color of the input object |
| query_material | *object* | *entry* | Return the material of the input object |
| query_size | *object* | *entry* | Return the size of the input object |
| query_shape | *object* | *entry* | Return the shape of the input object |

Table 3: Query modules of the program executor.

| Module | Input type | Output type | Description |
|---|---|---|---|
| relate_front | *object* | *scene* | Return all objects in front |
| relate_behind | *object* | *scene* | Return all objects behind |
| relate_left | *object* | *scene* | Return all objects to the left |
| relate_right | *object* | *scene* | Return all objects to the right |
| same_color | *object* | *scene* | Return all objects of the same color |
| same_material | *object* | *scene* | Return all objects of the same material |
| same_shape | *object* | *scene* | Return all objects of the same shape |
| same_size | *object* | *scene* | Return all objects of the same size |

Table 4: Relation modules of the program executor.

| Module | Input type | Output type | Description |
|---|---|---|---|
| filter_color[blue] | *scene* | *scene* | Select all blue objects from the input scene |
| filter_color[brown] | *scene* | *scene* | Select all brown objects from the input scene |
| filter_color[cyan] | *scene* | *scene* | Select all cyan objects from the input scene |
| filter_color[gray] | *scene* | *scene* | Select all gray objects from the input scene |
| filter_color[green] | *scene* | *scene* | Select all green objects from the input scene |
| filter_color[purple] | *scene* | *scene* | Select all purple objects from the input scene |
| filter_color[red] | *scene* | *scene* | Select all red objects from the input scene |
| filter_color[yellow] | *scene* | *scene* | Select all yellow objects from the input scene |
| filter_material[metal] | *scene* | *scene* | Select all metal objects from the input scene |
| filter_material[rubber] | *scene* | *scene* | Select all rubber objects from the input scene |
| filter_shape[cube] | *scene* | *scene* | Select all cubes from the input scene |
| filter_shape[cylinder] | *scene* | *scene* | Select all cylinders from the input scene |
| filter_shape[sphere] | *scene* | *scene* | Select all spheres from the input scene |
| filter_size[large] | *scene* | *scene* | Select all large objects from the input scene |
| filter_size[small] | *scene* | *scene* | Select all small objects from the input scene |

Table 5: Filter modules of the program executor.

# C Running Examples

Image

Scene

| ID | Size | Shape | Material | Color | x | y | z |
|---|---|---|---|---|---|---|---|
| 1 | Large | Sphere | Rubber | Gray | -2.07 | 0.93 | 0.69 |
| 2 | Small | Cube | Metal | Gray | -0.39 | -3.19 | 0.34 |
| 3 | Large | Cylinder | Rubber | Gray | 0.88 | -2.51 | 0.70 |
| 4 | Large | Sphere | Metal | Red | -0.82 | -1.23 | 0.70 |
| 5 | Small | Sphere | Metal | Red | -3.12 | -0.30 | 0.34 |
| 6 | Small | Cube | Rubber | Yellow | -1.41 | 2.57 | 0.34 |

Objects

Question: There is a gray ball that is the same size as the cylinder; what is it made of?

| Program | Output |
|---|---|
| scene | [1, 2, 3, 4, 5, 6] (list of object indices) |
| filter_shape[cylinder] | [3] |
| unique | 3 (single object) |
| same_size | [1, 4] |
| filter_color[gray] | [1] |
| filter_shape[sphere] | [1] |
| unique | 1 |
| query_material | rubber |

Answer: rubber

Figure 1: Running example of NS-VQA. Intermediate outputs from the program execution trace can be a scene (a list of objects), a single object, or an entry of certain attribute (i.e. "blue", "rubber").

Image

Scene

| ID | Size | Shape | Material | Color | x | y | z |
|---|---|---|---|---|---|---|---|
| 1 | Large | Cylinder | Rubber | Brown | -2.48 | 0.18 | 0.69 |
| 2 | Large | Cube | Rubber | Gray | -0.52 | 2.56 | 0.70 |
| 3 | Small | Cube | Metal | Gray | 1.88 | 2.02 | 0.35 |
| 4 | Small | Cylinder | Rubber | Green | -1.95 | -1.40 | 0.34 |
| 5 | Large | Sphere | Metal | Purple | 0.97 | -1.82 | 0.70 |

Objects

Question: Is the purple thing the same shape as the large gray rubber thing?

| Program | Output |
|---|---|
| scene | [1, 2, 3, 4, 5] |
| filter_size[large] | [1, 2, 5] |
| filter_color[gray] | [2] |
| filter_material[rubber] | [2] |
| unique | 2 |
| query_shape | cube |
| scene | [1, 2, 3, 4, 5] |
| filter_color[purple] | [5] |
| unique | 5 |
| query_shape | sphere    cube |
| equal_shape | no |

Answer: no

Figure 2: Running example of NS-VQA. Dashed arrow indicates joining outputs from previous program modules, which are sent to the next module.

Image

Scene

| ID | Size | Shape | Material | Color | x | y | z |
|---|---|---|---|---|---|---|---|
| 1 | Large | Cube | Metal | Cube | -3.24 | 0.55 | 0.69 |
| 2 | Small | Cube | Rubber | Brown | -0.52 | 3.88 | 0.34 |
| 3 | Large | Sphere | Rubber | Gray | 2.20 | -0.25 | 0.70 |
| 4 | Large | Cube | Metal | Gray | 0.69 | 1.93 | 0.70 |
| 5 | Large | Cylinder | Metal | Purple | 0.23 | -2.55 | 0.70 |
| 6 | Small | Cylinder | Metal | Yellow | -2.21 | -0.74 | 0.34 |

Objects

Question: How many large things are either purple cylinders or cyan metal objects?

| Program | Output |
|---|---|
| scene | [1, 2, 3, 4, 5, 6] |
| filter_color[cyan] | [] |
| filter_material[metal] | [] |
| scene | [1, 2, 3, 4, 5, 6] |
| filter_color[purple] | [5] |
| filter_shape[cylinder] | [5]   [] |
| union | [5] |
| filter_size[large] | [5] |
| count | 1 |

Answer: 1

Figure 3: Running example of NS-VQA.

Image

Scene

| ID | Size | Shape | Material | Color | x | y | z |
|----|-------|----------|----------|--------|-------|-------|------|
| 1 | Small | Cube | Metal | Brown | 0.95 | 0.18 | 0.35 |
| 2 | Large | Cylinder | Rubber | Cyan | -0.70 | 2.39 | 0.70 |
| 3 | Large | Cube | Metal | Green | 0.04 | -3.50 | 0.70 |
| 4 | Large | Cube | Metal | Yellow | -2.86 | -0.58 | 0.70 |

Objects

Question: There is a thing that is in front of the brown thing; what is its color?

| Program (Ours) | Output | | Program (Ground truth) | Output |
|----------------|--------|---|------------------------|--------|
| scene | [1, 2, 3, 4, 5, 6] | | scene | [1, 2, 3, 4, 5, 6] |
| scene | [1, 2, 3, 4, 5, 6] | | filter_color[brown] | [1] |
| filter_color[brown] | [1] | | unique | 1 |
| unique | 1 | | relate[front] | [2] |
| relate[front] | [2]  [1, 2, 3, 4, 5, 6] | | unique | 2 |
| intersect | [2] | | query_color | cyan |
| unique | 2 | | | |
| query_color | cyan | | | |

Answer (Ground truth): cyan

Answer (Ours): cyan

Figure 4: Running example of NS-VQA. Fail case: a spurious program leads to the correct answer. As compared to the ground truth, the spurious program predicted by our model does not significantly deviate from the underlying logic, but adds extra degenerate structures.

# D   Scene Parsing on Real Images

Input Image                                              Parsed Scene

Figure 5: Scene parsing results on real images. We handcraft real world CLEVR objects with paper boxes and rolls that are not well aligned with the synthetic scenes. We apply scene-parsing on the real objects without fine-tuning. Our model detects and extracts attributes from most objects correctly; in some cases, it mistakenly treats shadows as objects.