[Reviews · NeurIPS 2018]

Reviewer 1



The paper proposes a methodology that combines predefined structural information, such as position, shape, size, color, with symbolic-reasoning. Demonstrating a state-of-the-art performance over CLEVER dataset, with relatively small dataset, also evaluating their reasoning model for new Minecraft world setup. The paper is primarily an applications paper and in that regard, the experiments are well-executed and provide evidence for the claims in the paper. Clarity/Quality: The paper is well-written. contributions clearly laid out. Originality: The paper inspired and closely follows work of IEP, extending it with extracted structural information. Significance: The proposed approach demonstrates clear improvements: improving answer accuracy, generating plausible programs, reducing the need of labeled data, however, require stronger supervision with structure information. I believe structure supervision should be easier to achieve than annotated programs, therefore I’m in favor of this method. Some questions: • Reviewer is no expert in the specific methodology, while the structure is well-reasoned throughout, I wasn’t convinced why your approach generates better programs compared to IEP. Both use the same LSTM network. Can you elaborate? • I wonder if there a reason to compare your generated programs to IEP instead of SOTA TbD+reg+hres_ • Have you done ablation analysis, can you explain your design choices. For instance, L144, have you tried weighted version attention as well? After author's response: Thanks for clearing out my concerns.

Reviewer 2



This paper uses neural networks to parse visual scenes and language queries, transforming them into a logical representation that can be used to compute the output of the query on the scene. The logical representation is learned via a combination of direct supervision via a small number of traces and fine-tuning using end-to-end reinforcement learning. Advantages of the approach over existing approaches include: Reduction in the number of training examples, a more interpretable inference process and substantially increased accuracy. The overall approach shows great promise in increasing the performance of neural architectures by incorporating a symbolic component, as well as making them more robust, interpretable and debuggable. So I think this is a good direction for AI research to go in. The two approaches (neural and symbolic) complement each other nicely. The paper is also well-written, researched and executed so congratulations to the authors! Here are some possible avenues for improvement: - What are the flaws/shortcomings of this approach? Discussing this both theoretically and in terms of the actual obstacles the authors encountered while doing the research for this paper would be very valuable and a good addition to the paper. - Also I'd like to get some more insights about what are good techniques to try in this space. While well executed, the authors focus on one architecture that works well, without too much insight into why it works and what would work for harder benchmarks too. - As a minor comment, the authors claim that this approach can be useful to defend against adversarial attacks (line 57). On a high level, this seems to make sense, but nonetheless I would be very careful about this statement without sufficient research to back it up. It's probably safer to tone down the language some more here. Again good job and thanks for writing the paper! After having read the other reviews and the author response, my opinion that this paper should be published in NIPS with the feedback incorporated into the paper.

Reviewer 3



The paper studies the combination of deep representation learning for visual recognition and language understanding, and symbolic program execution for reasoning. The focus is on the development of a Visual Question Answering System, where question related to a scene are posted to the system. The idea is quite simple to understand: the system first recovers information related to the structure of the scene and represents it symbolically, together with the symbolic representation (in the form of a program) of the question. Then, the program is executed over the representation of the scene and the answer is computed. The first step is done by means of machine learning techniques for visual recognition and NLP techniques (when the question is posed in NL) to obtain the program associated to the question. The learned knowledge is then represented symbolically which provides the ability of a higher level of reasoning about the scene. The authors then show experimentally that the proposed system has a high percentage of accuracy on a benchmark dataset. They also show that it is more efficient in data and memory usage (the symbolic abstraction helps in reasoning and learning with fewer examples, as humans do). The authors state that to the best of their knowledge this is the first work that makes this combination allowing also for transparency and explanations in the reasoning process. To the best of my knowledge this true, but as this is not my area of expertise cannot be completely sure. The idea is novel, is timely, and sound to the extent I checked. The authors hypothesis is that machine learning techniques is not enough to perform high level reasoning tasks (as question answering) over visual scenes. A way to incorporate and structure context knowledge is necessary. This proposal achieves exactly that. My only concern is that the representation language used is not clearly defined. There is no formal definition or specification of what is a symbolic program. What language is used? What is a symbolic module? I this sense I find the work still behind of what symbolic reasoning is supposed to involve. Author's Feedback: the authors have answered satisfactorily my concerns.